



# Challenges and solutions in determining dilution ratios and emission factors from chase measurements of passenger vehicles

Ville Leinonen[1], Miska Olin[2], Sampsa Martikainen[2], Panu Karjalainen[2], and Santtu Mikkonen[1,3]

[1]Department of Technical Physics, University of Eastern Finland, Kuopio, Finland.
[2]Aerosol Physics Laboratory, Tampere University, Tampere, Finland.
[3]Department of Environmental and Biological Sciences, University of Eastern Finland, Kuopio, Finland.

*Correspondence to*: Ville Leinonen (ville.j.leinonen@uef.fi)

**Abstract.** Vehicle chase measurements used for studying real-world emissions apply various methods for calculating emission factors. Currently available methods are typically based on the dilution of emitted carbon dioxide ($CO_2$) and the assumption that other emitted pollutants dilute similarly. A problem with the current methods arises when the studied vehicle is not producing $CO_2$, e.g., during engine motoring events, such as on downhills. This problem is also encountered when studying non-exhaust emissions, e.g., from electric vehicles. In this study, we compare multiple methods previously applied for

determining the dilution ratios. Additionally, we present a method applying Multivariate Adaptive Regression Splines and a new method called Near-Wake Dilution. We show that emission factors calculated with both methods are in line with the current methods with vehicles producing $CO_2$. In downhill sections, the new methods were more robust to low $CO_2$ concentrations than some of the current methods. The methods introduced in this study can hence be applied in chase measurements with changing driving conditions and be possibly extended to estimate non-exhaust emissions in the future.

## 1 Introduction

Anthropogenically emitted gaseous compounds and particulate matter have effects on both climate and human health (Forster et al., 2021; Lelieveld et al., 2015). Vehicle emissions contribute to a significant proportion of those emissions, especially in urban environments. Vehicle emissions are regulated in legislation but the regulation even for new vehicles is still under development and the new regulations do not completely cover the existing fleet. Fulfilling the regulation requirements is

controlled in periodical technical inspection (PTI) of vehicles but the inspection protocol is limited to a few parameters, and, for example, the particle number (PN) is accounted only in some forerunner countries. Additionally, regarding particle emission regulations, only a fraction of the total emission is regulated. The limits for PN only consider nonvolatile particles, and the particle mass (PM) formed from the precursor gases via nucleation and condensation as the exhaust gas dilutes and cools upon exiting the tailpipe is mostly neglected. The amount of particle matter (both in terms of PN and PM) formed this

way can be considerable (Karjalainen et al., 2014b; Keskinen and Rönkkö, 2010; Kittelson, 1998; Giechaskiel et al., 2007).



A variety of measurement methodologies exist for studying emissions: official type-approval tests (that depend on the local legislation) are typically conducted by driving a predetermined driving cycle on a chassis dynamometer. In Europe, Portable Emission Measurement System (PEMS) protocol is also included for in-use compliance testing since 2016 (European Commission, 2016) including NOx, PN, and CO emissions in real drive. NOx emissions must be measured on all Euro 6 vehicles—passenger cars and light-commercial vehicles. On-road PN emissions are to be measured on all Euro 6 vehicles which have a PN limit set (diesel and GDI). CO emissions also must be measured and recorded on all Euro 6 vehicles. RDE emission limits (Emission Standards: Europe: Cars and Light Trucks: RDE Testing, 2023) are defined by multiplying the respective emission limit by a conformity factor (CF) for a given emission.

Remote sensing methods, such as snapshot measurements in fixed locations, or chasing vehicles with a mobile measurement unit sampling the diluted exhaust aerosol, are used for academic purposes (Karjalainen et al., 2014a; Simonen et al., 2019; Wang et al., 2010; Ježek et al., 2015b; Herndon et al., 2005; Shorter et al., 2005; Wang et al., 2017; Park et al., 2011; Pirjola et al., 2004). These methods have potential for elaborate use and could also be applied in monitoring vehicles fulfilling the regulation requirements.

The chase method has the considerable advantage of subjecting the exhaust aerosol to a real atmospheric dilution. The advantage of chase method is that the measured aerosol corresponds to the actual emission of the vehicle and not only a fraction (e.g., primary emissions only); however, the prevailing ambient conditions can strongly affect the particle formation, which is simultaneously an asset but also a drawback. On one side, this is the real particle population that is formed at a given time causing the immediate air quality effects, but on the other side, the method is hence not very repeatable between different testing conditions with respect to semi-volatile particle number and size. Additionally, the chase method is fast, and the individual measurement of vehicle's emission factor could be carried out in a minute time scales (Olin et al., 2023).

There exist several methods for calculating an emission factor (EF) from chase measurements (Hansen and Rosen, 1990; Zavala et al., 2006; Wihersaari et al., 2020; Ježek et al., 2015a). These methods are based on the $CO_2$ produced by the engine and on the assumption that all emitted components dilute similarly to $CO_2$. Downhill is problematic since engines do not generally inject fuel there because of no need for providing mechanical power (called engine motoring), and hence do not emit $CO_2$. However, previous studies (Rönkkö et al., 2014; Karjalainen et al., 2014a, 2016) suggest that engine motoring events can emit nanoparticles, originating from the lubricating oil. The chase vehicle observes these elevated concentrations in the plume, but it is difficult to assess the EF of the vehicle under measurement since the dilution ratio (DR) calculated with $CO_2$ is not available. In addition, most of the current methods have been used for a longer time interval, whereas short time interval EFs of accelerating and braking might be more interesting for studying. Also, a specific need to calculate EFs without $CO_2$ emissions is when studying non-exhaust emissions (e.g., particulate emissions from tires and brakes). In the future, when the fraction of



electric passenger vehicles is increasing, the research interest might shift towards non-exhaust emission. The new methods introduced in this study could be useful for estimating non-exhaust emission factors as well.

In this study, we will compare different calculation methods for EFs of vehicles based on chase measurements: particle number concentration (N) to $CO_2$ concentration ratio -based methods (Hansen and Rosen, 1990; Zavala et al., 2006; Olin et al., 2023),
a method that calculates the raw particle number concentration, $N_{raw}$, based on DR (Wihersaari et al., 2020), and two new methods to be introduced in this paper, based on Near-Wake Dilution and Multivariate Adaptive Regression Splines for DR in a remote-sensing-type chase measurement setting. Most of the methods used in this study can also be applied for snapshot-type measurements where DR needs to be defined. Our aim is to improve EF calculation, especially for short time intervals with variable DR, by achieving better understanding about the variables that affect DR. The new methods are both based on
the DR-modelling approach: using the DR calculation of the $CO_2$-based methods for the time periods when they work properly. We then extend the models to the whole measurement period by either using physical method (Near-Wake Dilution) or statistical method (Multivariate Adaptive Regression Splines) to estimate the DR for all measurement time points. We then compare the results from the new methods to the current methods for longer time intervals and separately for downhill sections. We also calculate DR and EF using only data from remote sensing measurements, without additional information on the
measured vehicle, such as on-board diagnostics (OBD) data (i.e., from the chase measurements). Development of this kind of methods are crucial if remote sensing measurements are applied on on-road monitoring of vehicle emissions, as suggested by, e.g., the European H2020-project CARES (https://cares-project.eu/).

## 2 Methods

### 2.1 Experiments

Particle number concentrations and $CO_2$ concentrations in exhaust plumes of six passenger vehicles (three diesel and three gasoline) were measured with the chase method during wintertime, in February in Siilinjärvi, Finland (Figure 1). The time and the location were selected because the main purpose of the measurement campaign was in studying wintertime real-world vehicle emissions, which is in the scope of future studies, applying methods introduced in this publication. The measurement instruments were installed inside the mobile laboratory of Tampere University (Aerosol and Trace gas Mobile Laboratory,
ATMo-Lab, Simonen et al. (2019); Rönkkö et al, (2017)). Data from the OBD and GPS from the test vehicles were saved at a 1 second time resolution (Figure 2). The chase route 1) was 13.8 km long including uphill and downhill driving, stops, accelerations and steady drive, also artificial short stops to simulate traffic lights. The route selection was based on bearing two major principles in mind. On one hand, there was a fuel station as a starting point with enough space for parking the test vehicles overnight, connection to electric grid to be used with electrical preheaters, and spaces were regularly cleared of snow.
On the other hand, the station was located close to roads ideal for tests: they were in good condition and were maintained well





during winter, and the traffic rates were very low implying that the background exhaust plumes are negligible. The route was also well suited for this study, because it included steep and long downhill sections.

The test protocol included a short period of engine idling at the beginning, driving the route, two predetermined stops and finishing the route at the start location. The time stamps of passing vehicles and other possible external emission sources were

recorded during the drives.

Information about the vehicles, individual drives and outside temperatures are shown in Table 1. During the test period of four days, the outside temperature varied between -9 and -28 ℃. The fleet included three (Euro 5-6) diesel vehicles (two passenger cars and one van) and three (Euro 6) gasoline vehicles (passenger cars). The number of measured drives totaled 33; in addition, there was a drive for every measurement day for measuring ambient background concentrations along the route. 11 drives

were dedicated to subfreezing–cold start (cold start in subfreezing temperatures) measurements, 12 to preheated–cold start measurements (using electric preheaters or fuel-combusting auxiliary heaters), and 10 to hot start measurements (the engine had been heated to its normal operating temperature by driving).

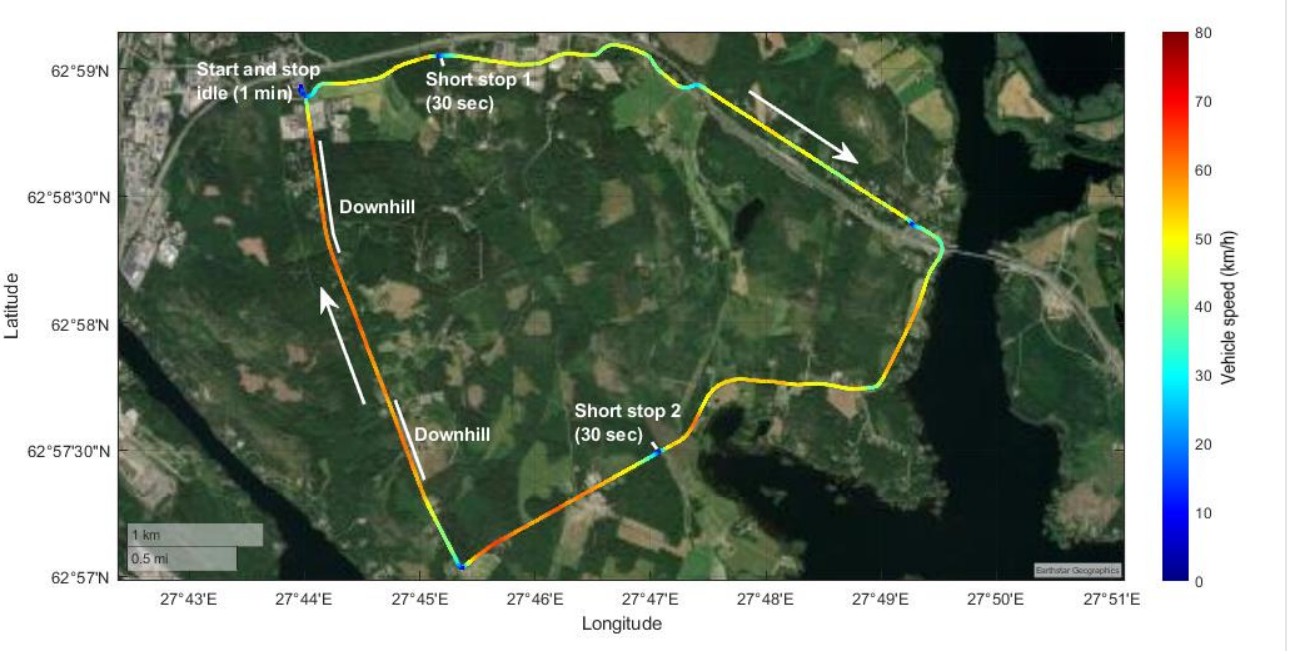

**Figure 1. Driving route consisting of low-traffic small roads in Siilinjärvi, Finland. The colored line indicates an example drive with the speed profile (color). Start and stop locations were the same position on a parking lot of a local fuel station. Two artificial short**




**stops were introduced along the test route to simulate traffic lights. Downhills that are used in the results section are indicated by white lines on the side of the route marking. Source: Earthstar Geographics.**

**2.2 Measurement setup**

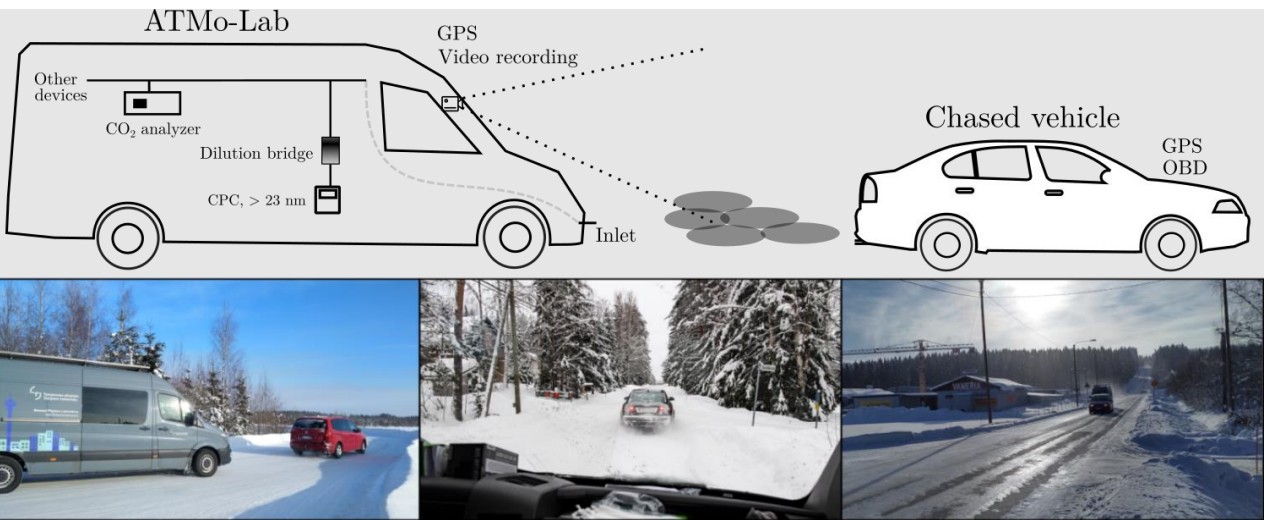


**Figure 2. Schematic view of the measurement setup used in this study and example photos from the chasing route for illustration of the chasing measurements. In addition, other devices were installed but their data were not used in this study.**

The measurement setup, including only the devices of which data are involved in this study, is presented in Figure 2. The number concentration of particles larger than 23 nanometers in diameter (N) was measured with an Airmodus model A23

condensation particle counter (CPC) and the $CO_2$ concentration with a LI-COR LI-840A analyzer. The exhaust sample was drawn to the instruments through a sampling inlet installed on the front bumper of the vehicle. Before the CPC, the sample was diluted with a set of bifurcated flow diluters (DR=158±14). The drives were also recorded with a video camera installed on the windshield and the location of the ATMo-Lab was recorded using GPS. OBD data from the chased vehicle were collected using OBDLink LX Bluetooth device (OBDLink® LX - Top-Notch Scan Tool Compatible With Motoscan, 2023).

All the devices were recording data with one second time resolution, which was averaged to the time resolution five second. Averaging makes the data more robust to small (1-2 sec) time differences between measurements from the vehicle (OBD) and variables measured with ATMo-Lab.

**Table 1: Information on the studied vehicles.**

| Vehicle | Fuel | Registration year | Engine displacement ($l$) | Emission class | Odometer reading ($km$) | Number of drives |
|---------|------|-------------------|---------------------------|----------------|-------------------------|------------------|





| Audi A6 | Diesel | 2008 | 3.0 | Euro 5 | 236,000 | 6 |
|---|---|---|---|---|---|---|
| Seat Alhambra | Diesel | 2012 | 2.0 | Euro 5 | 169,000 | 6 |
| VW Transporter | Diesel | 2019 | 2.0 | Euro 6 | 36,000 | 4 |
| Ford Focus | Gasoline | 2018 | 1.0 | Euro 6 | 78,000 | 5 |
| Skoda Octavia 1.0 | Gasoline (MHEV) | 2020 | 1.0 | Euro 6 | 1,000 | 6 |
| Skoda Octavia 2.0 | Gasoline | 2019 | 2.0 | Euro 6 | 21,000 | 6 |

## 2.3 Methods for calculating EF

The methods we use are mostly modeling DR and observed differences between measured and background concentrations and based on those calculating the EF of a vehicle. Used methods (introduced more in detail in the following subsections) for calculating EF can be divided into four categories based on whether the OBD data is used in the method and whether the method needs some additional (hereafter learning) data from other vehicles to evaluate the effect of some variables (e.g., speed change) to the emissions. Table 2 shows all the methods used in this study. All methods are introduced in the following subsections 2.3.1-2.3.7. Table 3 summarizes the main differences of the methods described in subsections 2.3.1-2.3.7 and shows the formulas used to calculate the EF in each of the methods.

The dataset used in this study was limited to considering only times when the chased vehicle was moving, i.e., its speed was positive. Also, the effect of chase distance, i.e., the distance between the chased vehicle and ATMo-Lab, was assumed to be constant, not affecting the dilution ratio of emission.

**Table 2: Division of the methods for calculating EF of a vehicle. OBD data means the data collected from the chased vehicle (see also Figure 2) and learning data means the data collected from other drives of the same vehicle and from other vehicles (including data from ATMo-Lab and, also from OBD if its data is used). Methods are introduced in more detail in subsections 2.3.1-2.3.7.**

| | | Uses learning data | |
|---|---|---|---|
| | | yes | no |
| Uses OBD data | yes | MARS-OBD, NWD | $N/CO_2$ integral, $N/CO_2$ linear, $N/CO_2$ RRPA, $N_{raw}$ |





| | | |
|---|---|---|
| no | MARS-chase | N/CO$_2$ Traficom |

Methods that require data to be fitted before applying into DR estimation or EF calculation were fitted using DR calculated from N$_{raw}$ method as a response variable. Only the data from the times with exhaust mass flow rates (Q) exceeding 0.3 $g\ s^{-1}$ and fuel flow rates exceeding 0.02 $g\ s^{-1}$ were used in forming models, which were then used for the whole data also including the times with the flow rates below those limits.

Other methods of modeling DR (NWD and MARS-methods, described below) are based on the observed linear or non-linear
dependencies between DR and explanatory variable(s). These methods assume that the factors affecting DR measured in the situations where the measured vehicle is not in the engine motoring mode can be extrapolated also to situations with the motoring mode. Hence, for the downhill sections, the following methods do not calculate the DR based on the measured CO$_2$; instead, they use other parameters not based on CO$_2$ (some examples include vehicle speed $v_t$, exhaust flow rate $Q$ and the vehicle rear shape) to estimate the DR.

For calculating EF and its uncertainty, bootstrap sampling (Efron, 1979) has been used to estimate the uncertainty in EF calculations. A bootstrap sample is a random sample of observations (observation = time point) with replacement, i.e., one observation can occur multiple times in a bootstrap sample. The analysis, e.g., fitting the model and calculating the EF is performed for this bootstrap sample. Multiple bootstrap samples are usually taken, here 100 is the number of samples.

Bootstrap helps to estimate the whole uncertainty, in this case the uncertainty related to e.g., differences in vehicle driving
profile, possible uncertainties in time allocation, and uncertainty in model fitting. Bootstrap is useful when estimating complex estimators or their uncertainty, without (here) explicitly estimating uncertainties of single sources of uncertainty and covariance structure of uncertainties.

**Table 3: Summary of the methods used in this study. Formulas to calculate EF, main differences to other methods, and references to the literature describing the method. Methods are introduced in more detail in subsections 2.3.1-2.3.7.**




Atmospheric
Measurement
Techniques

Discussions

| Method | Formula to calculate EF | Differences to other methods | Reference |
|---|---|---|---|
| **Methods using fraction of N and CO2** | | | |
| N/CO$_2$ integral (subsection 2.3.1) | $$\frac{\int_t [N_t^{meas} - N^{bg}]dt}{\int_t [CO_{2,t}^{meas} - CO_2^{bg}]dt} * \frac{a_{\frac{g}{cm^3}}^{ppm} * a_{g_{fuel}}^{g_{CO_2}} * m_{fuel}}{s_{drive}}$$ | $m_{fuel}$ is taken from OBD measurements of the vehicle. For other terms of the formula, see subsection 2.3.1. | Hansen and Rosen (1990) |
| N/CO$_2$ Traficom (2.3.2) | $$\frac{\int_t [N_t^{meas} - N^{bg}]dt}{\int_t [CO_{2,t}^{meas} - CO_2^{bg}]dt} * \frac{a_{\frac{g}{cm^3}}^{ppm} * a_{g_{fuel}}^{g_{CO_2}} * m_{fuel}}{s_{drive}}$$ | $m_{fuel}$ is taken from Finnish national database for vehicles (Traficom), otherwise as N/CO$_2$ integral. | This study |
| N/CO$_2$ linear (2.3.3) | $$\frac{\Delta N}{\Delta CO_{2\ linear}} * \frac{a_{\frac{g}{cm^3}}^{ppm} * a_{g_{fuel}}^{g_{CO_2}} * m_{fuel}}{s_{drive}}$$ | Ratio of $N$ and $CO_2$ ($\frac{\Delta N}{\Delta CO_{2\ linear}}$) is estimated using the linear model to the background corrected values of $N$ and $CO_2$. Otherwise as N/CO2 integral. | Zavala et al. (2006) |
| N/CO$_2$ RRPA (2.3.4) | $$\frac{\Delta N}{\Delta CO_{2\ RRPA}} * \frac{a_{\frac{g}{cm^3}}^{ppm} * a_{g_{fuel}}^{g_{CO_2}} * m_{fuel}}{s_{drive}}$$ | Ratio of $N$ and $CO_2$ ($\frac{\Delta N}{\Delta CO_{2\ RRPA}}$) is estimated using the robust linear model to the measured values of $N$ and $CO_2$ without background correction. Otherwise as N/CO$_2$ integral. | Olin et al. (2023) |
| **Methods using dilution ratio** | | | |
| $N_{raw}$ (2.3.5) | $$\frac{\int_t [(N_t^{meas} - N^{bg}) * DR_{N_{raw},t} * Q_t]dt}{\rho_{exh} * \int_t v_t \, dt}$$ | Dilution ratio ($DR_{N_{raw},t}$) is calculated based on measured dilution of $CO_2$. For other terms of the formula and details, see subsection 2.3.5. | Wihersaari et al. (2020) |





| Near-Wake Dilution (NWD, 2.3.6) | $$\frac{\int_t[(N_t^{meas} - N^{bg}) * DR_{NWD,t} * Q_t]dt}{\rho_{exh} * \int_t v_t\, dt}$$ | Dilution ratio $DR_{NWD,t}$ is calculated based on a linear function of the ratio of the vehicle speed $v_t$ and the mass exhaust flow rate $Q_t$. See subsection 2.3.6 and Supplement for more details. Otherwise as $N_{raw}$. | This study |
|---|---|---|---|
| MARS-OBD (2.3.7) | $$\frac{\int_t[(N_t^{meas} - N^{bg}) * DR_{MARS-OBD,t} * Q_t]dt}{\rho_{exh} * \int_t v_t\, dt}$$ | Dilution ratio $DR_{MARS-OBD,t}$ is calculated based on Multivariate Adaptive Regression Spline (MARS) model for DR. See subsections 2.3.7 and 3.2 for more details. Otherwise as $N_{raw}$. | This study |
| MARS-chase (2.3.7) | $$\frac{\int_t[(N_t^{meas} - N^{bg}) * DR_{MARS-chase,t} * Q_{MARS-chase,t}]dt}{\rho_{exh} * \int_t v_t\, dt}$$ | Dilution ratio $DR_{MARS-chase,t}$ is calculated based on Multivariate Adaptive Regression Spline (MARS) model for DR. Variables available from ATMo-Lab (i.e., no OBD data) are used. Also, the exhaust flow rate ($Q_{MARS-chase,t}$) is estimated using splines. See subsections 2.3.7 and 3.2 for more details. Otherwise as $N_{raw}$. | This study |

### 2.3.1 N/CO₂ integral

The simplest method to calculate EF is based on N/CO$_2$ measured from the diluted exhaust. The method was introduced by Hansen and Rosen (1990) and has been widely used thereafter. It is based on the relation of the excess CO$_2$ ($\Delta CO_2 = CO_{2,t}^{meas} -$

$CO_2^{bg}$) and particle concentration ($\Delta N = N_t^{meas} - N^{bg}$). Here the superscripts meas and bg denote measured and background concentrations, respectively. Here $t$ denotes that the measured concentrations have been measured specifically at time $t$, whereas the background concentrations have been defined as a median of the background measurement measured at the same route on the same day. However, the method by Hansen and Rosen, (1990) uses the following integral form (over a longer measurement period than, e.g., one second) to diminish possible uncertainties caused by imperfect time synchronizations of

the devices measuring CO$_2$ and the studied pollutant:



$$EF_{\Delta N/\Delta CO_2} = \frac{\int_t [N_t^{meas} - N^{bg}]dt}{\int_t [CO_{2,t}^{meas} - CO_2^{bg}]dt} * \frac{a_{\frac{g}{cm^3}}^{ppm} * a_{g_{fuel}}^{gCO_2} * m_{fuel}}{s_{drive}} \quad (1)$$

where $CO_2$ concentrations are in ppm and particle concentrations in 1 cm$^{-3}$. $a_{g/cm^3}^{ppm}$ is the conversion factor for $CO_2$ from ppm to g cm$^{-3}$ ($10^6/0.0018 = 5.55*10^8$, where $10^6$ is a number of molecules and 0.0018 is the approximate density of $CO_2$ [g cm$^{-3}$] at 20 °C), $a_{g_{fuel}}^{gCO_2}$ is the conversion factor for $g_{CO_2}$ to $g_{fuel}$ (2392/750 = 3.189 for gasoline and 2640/835 = 3.162 for diesel,

where the 2392 and 2640 are the approximate masses of CO2 produced [g] per liter of fuel for gasoline and diesel respectively (Conversion Guidelines-Greenhouse gas emissions, 2023) and 750 and 835 are the approximate densities [g] of a fuel [$g\ l^{-1}$] for gasoline and diesel, respectively. Those densities are within the ranges of densities provided by one major fuel supplier in Finland (Neste Futura 95E10 Technical Data Sheet, 2023; Neste Futura Diesel -29/-38 Technical Data Sheet, 2023)). Variable $m_{fuel}$ is the mass of the used fuel (in g, from OBD data) and $s_{drive}$ is the length of the drive (in km). In this study, EF is

calculated over the whole measurement period and EF is expressed in 1 km$^{-1}$. This method (and all other N/CO$_2$ method versions) is based on the assumptions that CO$_2$ and the pollutant dilute equally in an exhaust plume and that the amount of emitted CO$_2$ is directly related to the fuel consumption. Whereas the N/CO$_2$ integral method is robust to imperfect time synchronizations and to the engine motoring events (because the integral in the denominator never becomes very small, unlike in cases with, e.g., one-second resolution), the method, however, assumes also that EF is constant during the integration time

period in chase measurements (Olin et al., 2023).

### 2.3.2 N/CO₂ Traficom

The N/CO$_2$ Traficom method is calculated similarly to N/CO$_2$ integral method, over the whole measurement period, but the fuel consumption $m_{fuel}$ is estimated from the national vehicle database (Traficom) instead of using actual consumption from OBD. Using the fuel consumption estimation from the register plates makes the method independent from OBD-data, i.e., the

method can be calculated directly based on the measurement data from ATMo-Lab. This kind of a method, that is not using OBD-data, is suitable, e.g., for real-world emission monitoring approaches for private vehicles driving on public roads. We have used constant fuel consumptions reported for combined driving (combining urban and extra-urban driving) that are between 4.6 (Ford) and 7.6 (VW) liters of fuel per 100 km.

### 2.3.3 N/CO₂ linear

The N/CO$_2$ linear method used, e.g., by Zavala et al. (2006) was also tested in this study. The method estimates N/CO$_2$ by fitting a line for $\Delta N$ and $\Delta CO_2$. The slope of that line is used to replace the first fraction term in Eq. (1) when calculating EF. The used linear model has an assumption that the line passes the origin, i.e., with no emitted CO$_2$, no particles are emitted. Therefore, non-exhaust particles are not counted. This method also assumes that EF is constant during the time period used

for fitting. However, as the drives cannot be assumed to have constant EF due to multiple different sections of driving, the

linear model is fitted separately to one-minute time periods, in which the vehicle can be assumed to have more constant EF throughout the period. For the periods when the slope is estimated to be negative, the EF is set to zero.

### 2.3.4 N/CO₂ RRPA

The RRPA (Robust Regression Plume Analysis) method presented in (Olin et al., 2023) is based on the N/CO₂ linear method but without no need to determinethe background concentrations of N and CO₂ Similarly to N/CO₂ linear method, the slope is

used to replace the first fraction term in Eq. (1) when calculating EF.

Contrary to the N/CO₂ linear method, this method uses robust linear model (in this study using *rlm*-function in R environment (R Core Team, 2022)) for fitting the line. We used robust linear regression instead of ordinary least squares approach because the data contains varying number of datapoints which can be considered as outliers, in statistical point of view, and those could bias the fit for the slope in ordinary least squares estimation (Mikkonen et al., 2019). The robust regression automatically

downweighs the possible outliers by giving less weight to the data points that are not close to the estimated line. Hence, momentary disturbances (such as from other pollutant sources near the measurement location) should not disturb the estimation of the slope. As for the N/CO₂ linear method, the N/CO₂ RRPA method assumes constant EF for the fitted period and is also fitted to a one-minute time periods. For the periods when the slope is estimated to be negative, the EF is set to zero.

### 2.3.5 $N_{raw}$

A bit more advanced method (based on the method by Wihersaari et al. (2020)) to calculate DR and EF is using the measured and raw concentrations of CO₂ and using the exhaust mass flow rate (Q):

$$DR_{N_{raw},t} = \frac{CO_{2,t}^{raw} - CO_2^{bg}}{CO_{2,t}^{meas} - CO_2^{bg}} \quad (2)$$

$$EF_{N_{raw}} = \frac{\int_t [(N_t^{meas} - N^{bg}) * DR_{N_{raw},t} * Q_t] dt}{\rho_{exh} * \int_t v_t dt} \quad (3)$$

where $N_t^{meas}$ is the measured particle number concentration, $N^{bg}$ is the estimated background particle number concentration,

$CO_{2,t}^{raw}$ is the concentration of CO₂ in the raw exhaust (calculated from the OBD data), $\rho_{exh}$ is the exhaust density (air density at the standard temperature of 20 °C used here), and $v_t$ is the vehicle speed. We denote the method as N$_{raw}$ method from here onwards. This method can be thought of as the best performing model in a real-world chasing situation with varying EF and DR. However, this method requires well-synchronized data. Five second time resolution was used, as it is not so prone to errors caused, e.g., by engine motoring events.





### 2.3.6 Near-Wake Dilution (NWD)

In the NWD method, we are building a robust linear model for DR as a linear function of the ratio of the vehicle speed $v_t$ and the mass exhaust flow rate $Q$ , taking also into account the shape of the vehicle's rear and the fuel used. The method is based on the assumption that the outdoor air passing by the vehicle's rear while driving dilutes the exhaust plume and that the dilution is proportional to the ratio of the mass flows passing the rear and exhausted from the tailpipe (Chang et al., 2012) . The method minimizes the weighted linear model (iterated reweighted least squares robust regression):

$$DR_{NWD,t} = \gamma + \kappa \frac{v_t}{Q_t} \ (4)$$

where dilution ratio at time t ($DR_t$) used to fit the model is calculated from the OBD chase measurement data as in the $N_{raw}$ method (Eq. (2)). Parameters $\gamma$ and $\kappa$ are coefficients fitted for every vehicle measured in this study. More detailed derivation of the formula and detailed discussion about the possible variables that are related to the parameters $\gamma$ and $\kappa$ are presented in the Supplement. The NWD model is fitted separately for each vehicle, except when the data from the studied vehicle is not used to fit a model (Fig. 6). In that case, the rear shape has been used as a categorical variable for the five-vehicle data to fit the NWD model. Categorical variables $b_1$ and $b_2$ estimate the effect of different rear types on DR: $DR_{NWD,t} = \gamma + b_1 + (\kappa + b_2) \frac{v_t}{Q_t}$.

As the model is only dependent on the speed and exhaust flow, the model assumes that the distance from the vehicle remains constant and is independent of the speed (the effect of the distance is incorporated into the kappa and gamma parameters). Constant driving distances were tried to maintain during these chase measurements. DR is calculated for all datapoints using the modeled dependency (presented later in Fig. 3).

EFs with the NWD method were then calculated similarly to the $N_{raw}$ method in formula (3), with different method to calculate the dilution ratio being the only difference between methods. The NWD method is robust to engine motoring events because the $CO_2$ concentration is not involved in the equation used to calculate EF (after fitting the kappa and gamma parameters). In addition, the method can possibly be used to determine non-exhaust emissions as well.

### 2.3.7 Multivariate Adaptive Regression Splines (MARS)

We used Multivariate Adaptive Regression Splines (MARS: Friedman, 1991; Hastie et al., 2009) to model the dependency of DR on certain variables that could affect the dilution of exhaust, I.e. vehicle exhaust flow rate, speed, speed change (acceleration), altitude change, and direction of wind.. Besides variables that are fitted with splines, two categorial variables describing the rear shape and fuel type used in the vehicle were used. Those categorical variables affect only the level, not the shape of the spline (see Fig. 4).



To avoid overfitting, i.e., that the model fits well to the learning data but is not generalizable to any new dataset, we used 5-fold cross-validation (Hastie et al., 2009). In 5-fold cross validation, the dataset is divided into five distinct subsets of the same

size. Then four of those subsets are used to train the model (training dataset) and one is used to test the fit of the model to new dataset (testing dataset). This is repeated five times, so that each subset is once used as a testing dataset.

We built two methods based on MARS: one is based on all variables (OBD-data and the data from chase measurement; a method called MARS-OBD), and the other one is based on the measured data consisting only variables that are available with remote sensing methods (a method called MARS-chase).

EFs from the MARS methods were calculated similarly to the $N_{raw}$ method (formula (3)), with the only difference to $N_{raw}$ in how the DR is calculated. As for NWD, DR is calculated for all datapoints using the modeled dependency (presented later in Fig. 4). MARS models are also robust for engine motoring events or even for non-exhaust emissions, like the NWD model, because the $CO_2$ concentration is not used (after the model construction). In addition, the MARS-chase model can be used in real-world emission monitoring approaches.

**3 Results and discussion**

**3.1 Fitting the NWD model parameters**

Our results indicate that DR can be approximated with linear function of the ratio of $v_t$ and Q; hence, it was used as one method to estimate DR. Figure 3 shows the robust linear regression fits between DR and $\frac{v_t}{Q}$.

According to the results, in addition to $\frac{v_t}{Q}$, we suppose that modelled DR is mostly affected by the rear shape of the vehicle

(included in the parameter $\kappa$). Generally, the values of $\frac{v_t}{Q}$ are higher for the gasoline vehicles compared to the diesel vehicles, due to lean burn combustion used in diesel engines. This results also in higher values of DR (determined with Eq. (2)) for the gasoline vehicles.



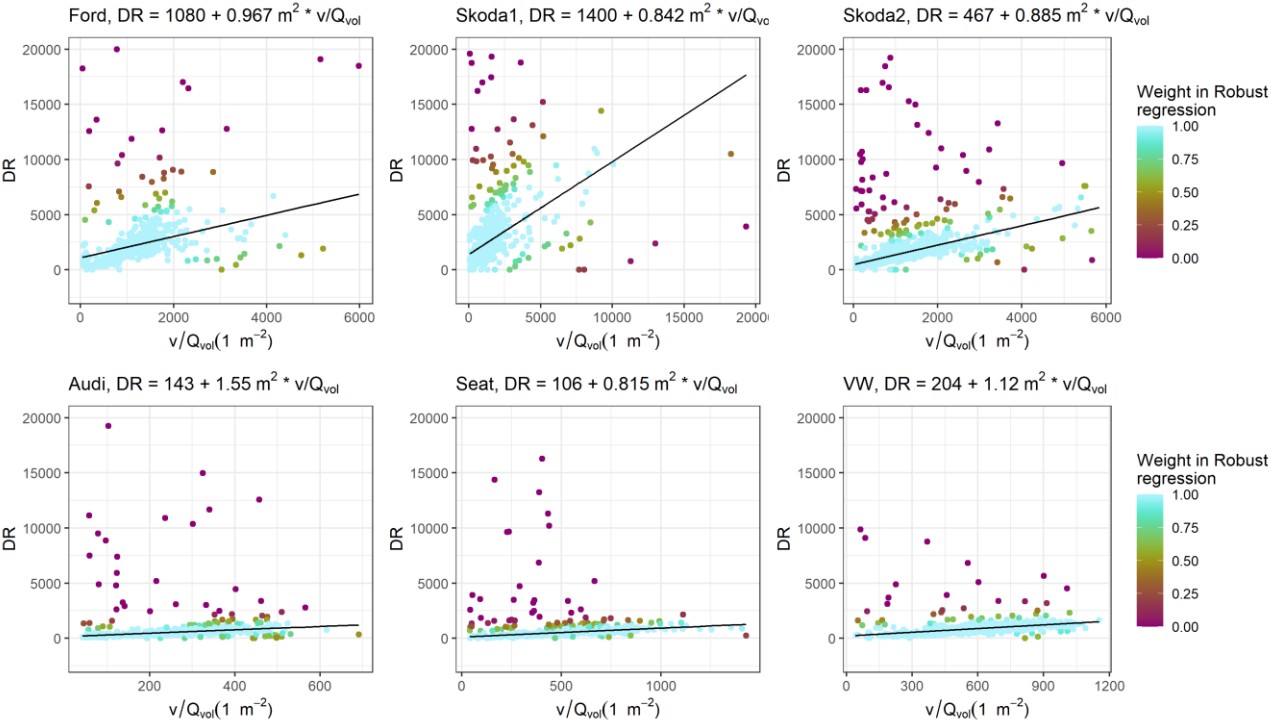

**Figure 3. Robust linear regression fits for DR for each vehicle used in NWD method. The color represents the weight of the observation in the final robust linear fit. The equations of the linear fits are shown in the titles of each subplot. Volumetric exhaust flow rate $Q_{vol} = \frac{Q}{3.6 \cdot \rho_{fuel}}$ has been used in this figure instead of the mass flow rate used elsewhere, because the NWD model is based on the volumetric flows.**

### 3.2 Constructing the MARS models

Figure 4 shows the behaviour of the splines in the measured data between DR and the predictor variables used in the MARS models. The shape of the splines is the same for all vehicles, as it is defined from the full dataset, but the level varies due to different properties of the vehicles, such as fuel and presumably the rear shapes.

The variables used in the models shown in Fig. 4 are organized so that the variables in the upper row are for the method using also the OBD-data from the chased vehicle (MARS-OBD) and the variables in the lower row are for the method using only variables from ATMo-Lab (MARS-chase). With the MARS-OBD method, changes in Q explain most of the changes observed in DR, and the dependency of Q on DR is as expected from the concept behind the NWD model. In addition to Q, wind component calculated abeam of the vehicle was seen to affect the DR, but the effect is very minor. Unlike in the MARS-chase method, variables such as speed change and altitude change were not needed (based on their effect on the model fit, measured





with R² values) in the MARS-OBD method, which indicates that the changes in Q (and slightly in the lateral wind speed) sufficiently explain most of the changes in DR.

For the MARS-chase method, the effect of Q was replaced by using several variables that could explain the power generated by an engine – and thus Q. The result seems to be in line with theory, the most evident changes to DR being caused by changes in driving speed (e.g., when accelerating) and altitude (e.g., when driving uphill), and the absolute speed of a vehicle (due to air drag). Observed dependencies of those variables with DR were described with piecewise linear splines with one or two threshold values (knots). The effect of changes in speed and altitude were close to linear. The effect of $v_t$ was not linear, as

the DR had its minimum after threshold speed slightly higher than 10 m/s.

**Figure 4. Multiple adaptive regression spline fits for logarithm (natural) of DR shown for each variable used in MARS-OBD (upper row) and in MARS-chase (lower row). Measurement points are colored based on the vehicle used. Different colored lines show the regression splines for each vehicle (see also categorical variables in the method description section 2.3.7), with some splines overlapping with each other.**





### 3.3 Comparison of the EF calculation methods for the whole drive

When the calculated DR estimates were applied on the EF calculation for the whole drive, it was seen that the results are mostly similar with all methods. Figure 5 illustrates how the calculated EF varies with different methods when applied on two different vehicles, one with gasoline and one with diesel engine, on two different drives with varying outside temperature.

The results in Figure 5 give confidence on EF calculation with varying information in use, as the methods with different background information end up mostly to within an order of magnitude. This is specifically good news for monitoring-type measurements, to be performed on-road, having limited information on the monitored vehicle. However, there can still be some notable differences between the methods, for example the difference of a factor of 2-3 between the $N_{raw}$ and other methods for Skoda2 –24 °C. The clearest anomalies from the consensus of EF are $N/CO_2$ RRPA for the Skoda2 –26 °C drive

being 25 to 45 % of the EFs given by other methods than $N_{raw}$ and $N/CO_2$ linear, and $N_{raw}$ method for both Skoda2 drives showing 2 to 4.2 times higher EFs than most of the methods (other than $N/CO_2$ linear and $N/CO_2$ RRPA). For RRPA some of the one-minute interval EFs were estimated to be zero, which probably explains the lower EFs calculated for that method. For $N_{raw}$ method, the difference comes from the time points where dilution ratio is estimated to be larger, e.g., in NWD and MARS-models, i.e., points clearly above modeled lines in Figs 3 and 4. If measured concentration of particles above background

$N_t^{meas} - N^{bg}$ is high enough for those points, it results also high EF for that point.



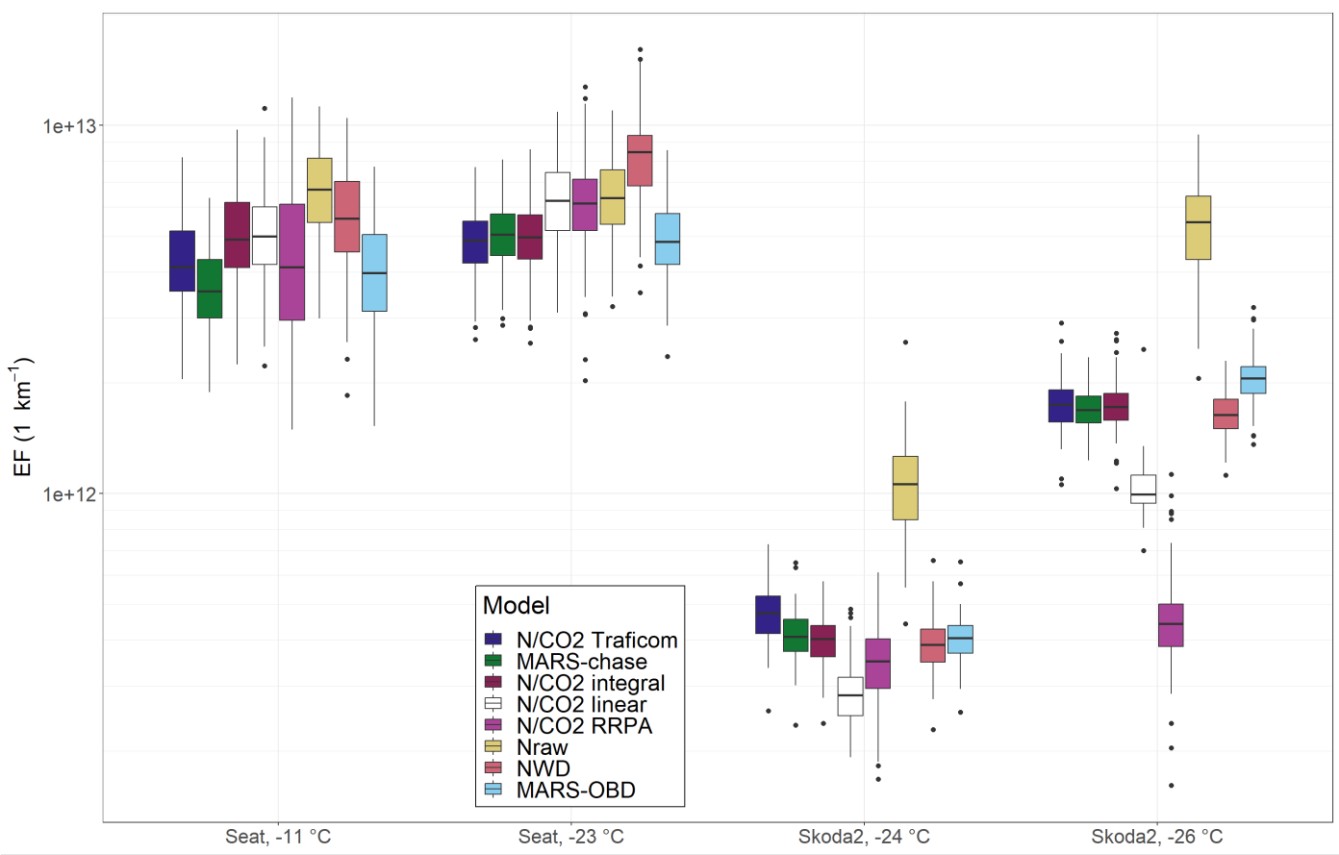

**Figure 5. Emission factor examples of > 23 nm particles for Seat and Skoda 2, hot starts, except Seat −11 C that is with the subfreezing–cold start, and Skoda −26 C that is with the preheated–cold start. Results are calculated from 100 bootstrap samples (see Sect. 2.3 for the description of bootstrap sampling).**

The methods that use learning data (the MARS methods and the NWD method, see Table 2) were validated with leave-one-out type cross-validation by omitting one of the vehicles from the model fitting and then by applying the fitted coefficients to predict the EFs for the omitted vehicle. The results are shown in Fig. 6, which confirms the findings in Fig. 5, that the constructed methods can calculate the EFs also for the vehicle omitted from the model construction. For the methods that don't use learning data (all $N/CO_2$ methods and the $N_{raw}$ method), i.e., data from the other drives to form a model, the results are almost the same (bootstrap sampling can change the calculated EFs slightly) as in Fig. 5. For Skoda2, the MARS-chase method shows higher EF values than the other methods in Fig. 6. This is probably because the data measured with Audi (being the only vehicle having a similar rear shape to Skoda2) has been used in the MARS-chase model to estimate the effect of the rear shape on the DR (see Sect. 2.3.7 for categorical variables and Fig. 3 for the fits). However, using the data from a diesel vehicle





in modelling DR for a gasoline vehicle may not work properly due to different dilution mechanics (as is also observed from the different fitting parameters obtained with using the NWD model)) even though the fuel type parameter for Skoda2 is different than for Audi. In addition, Audi is the only vehicle in this study having two exhaust pipes on both sides of the vehicle rear; therefore, the dilution mechanics may differ notably from the other vehicles. Thus, the rear shape parameter (constant categorial variable used to estimate the effect of the rear shape on DR) might have increased the estimated DR for Skoda2, and

hence also the estimated EF. One solution for this issue would be to increase the sample size of the vehicles, probably leading to a better estimate for the rear shape of Skoda2 in the MARS-chase method. For Seat, the MARS-chase method gives similar results to the other methods; however, the NWD method gives slightly higher EFs than the other methods. This is due to imperfect modeling of dilution ratio of Seat based on the model from other five vehicles. This indicates that EFs could be calculated in-situ based on the measurements from ATMo-Lab and OBD, if the OBD data is required in the method. The

increase in the number of vehicles in the learning data would probably increase the accuracy of the methods that require learning data, including the MARS-chase method as well.



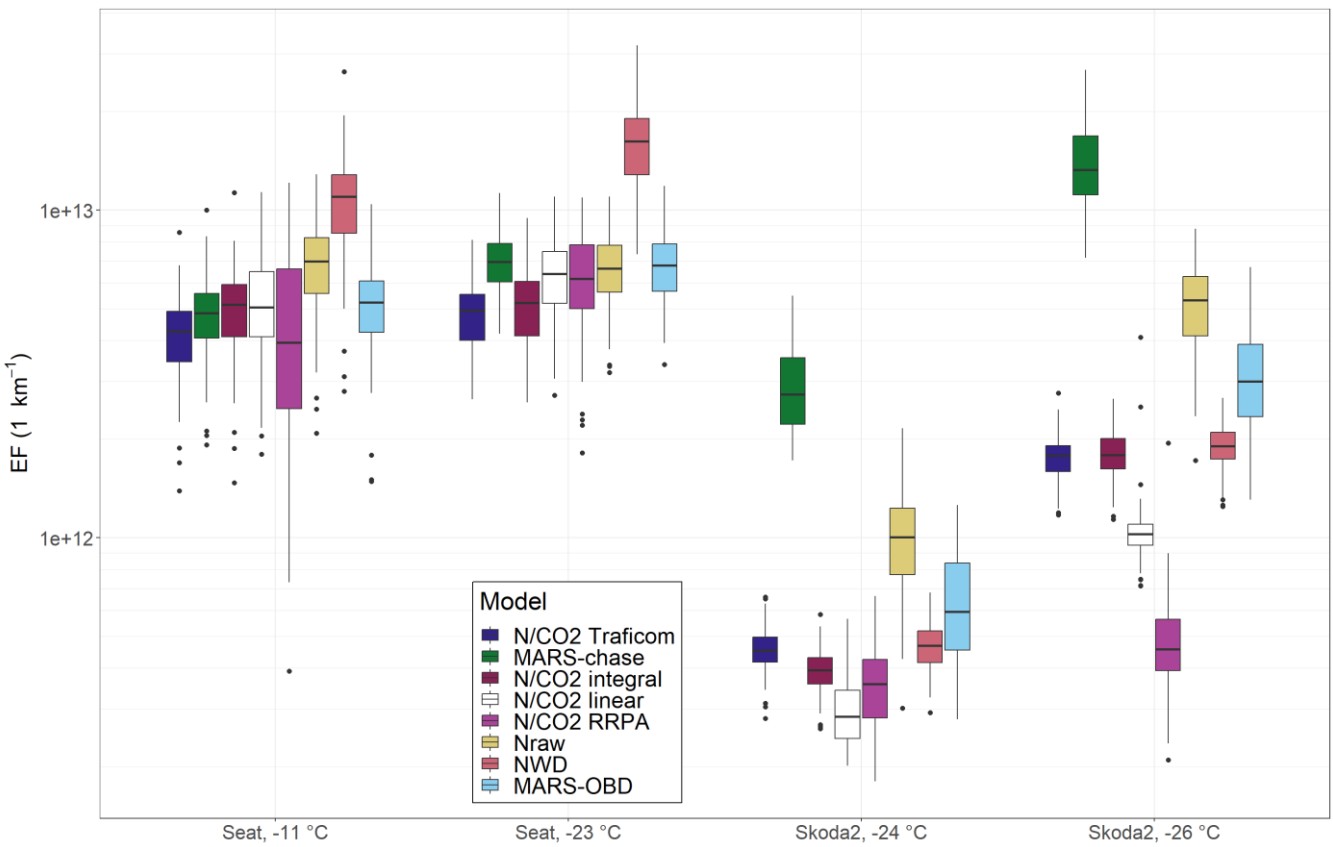

**Figure 6. Emission factors for the drives of which data are omitted (details in Sect. 3.3) from the model construction (MARS-chase, NWD, MARS-OBD) for Seat and Skoda2, hot starts, except Seat –11 C that is with the subfreezing–cold start, and Skoda –26 C that is with the preheated–cold start. Results are calculated from 100 bootstrap samples (see Sect. 2.3 for the description of bootstrap sampling).**

### 3.4 Comparison of the EF calculation methods for the downhill section

When examining how the different methods perform in different driving conditions, such as the change in the altitude, Fig. 7 shows that, overall, the methods agree quite well for Seat, but there are a lot of discrepancies for Skoda2. It is obvious why the $N/CO_2$ Traficom method overestimates the EFs during the downhill section: because the used fuel consumption refers to the combined driving fuel consumption data, i.e., to a much higher consumption than really occurs in downhills. In addition, the $N_{raw}$ method gives relatively high estimates for EFs, especially for Skoda2. This is due to relatively low $CO_2$ values observed at the times with high particle emissions, resulting in higher DRs with the $N_{raw}$ method compared to the other methods. $N/CO_2$ linear shows clearly lower EF values for Skoda2, similarly to, but less pronounced, in Figs. 5 and 6. For RRPA method, many of the EF estimates for bootstrap samples (89 out of 100 for Skoda2, -24 °C and 39 for Skoda2, -26 °C) are zero, i.e. for every minute interval (2 or 3 intervals in each bootstrap sample), the estimated linear dependency between N and $CO_2$ concentrations





is negative, and hence the EF is estimated to be zero. The assumption of constant EF is not valid in downhill sections, and the concentrations of N and $CO_2$, and exhaust flow rate are mostly lower than average of the whole round, whereas the DR, that is used in many other methods, is mostly higher than average of the whole round. We believe that those are the reasons why

RRPA is giving EFs so different than other methods for downhill sections.

Other methods (MARS-chase, N/$CO_2$ integral, NWD, and MARS-OBD) give similar values for EF. This is kind of expected as the methods are fitted using data from the full drives (as in the case in Figs. 5 and 7). Therefore, the N/$CO_2$ is estimated mostly from the data with above-zero fuel consumption; hence, the number of particles emitted per extra $CO_2$ emitted should be estimated well. The other methods are also based on data with above-zero fuel consumption; thus, the dilution ratio for the

downhills can also be estimated.

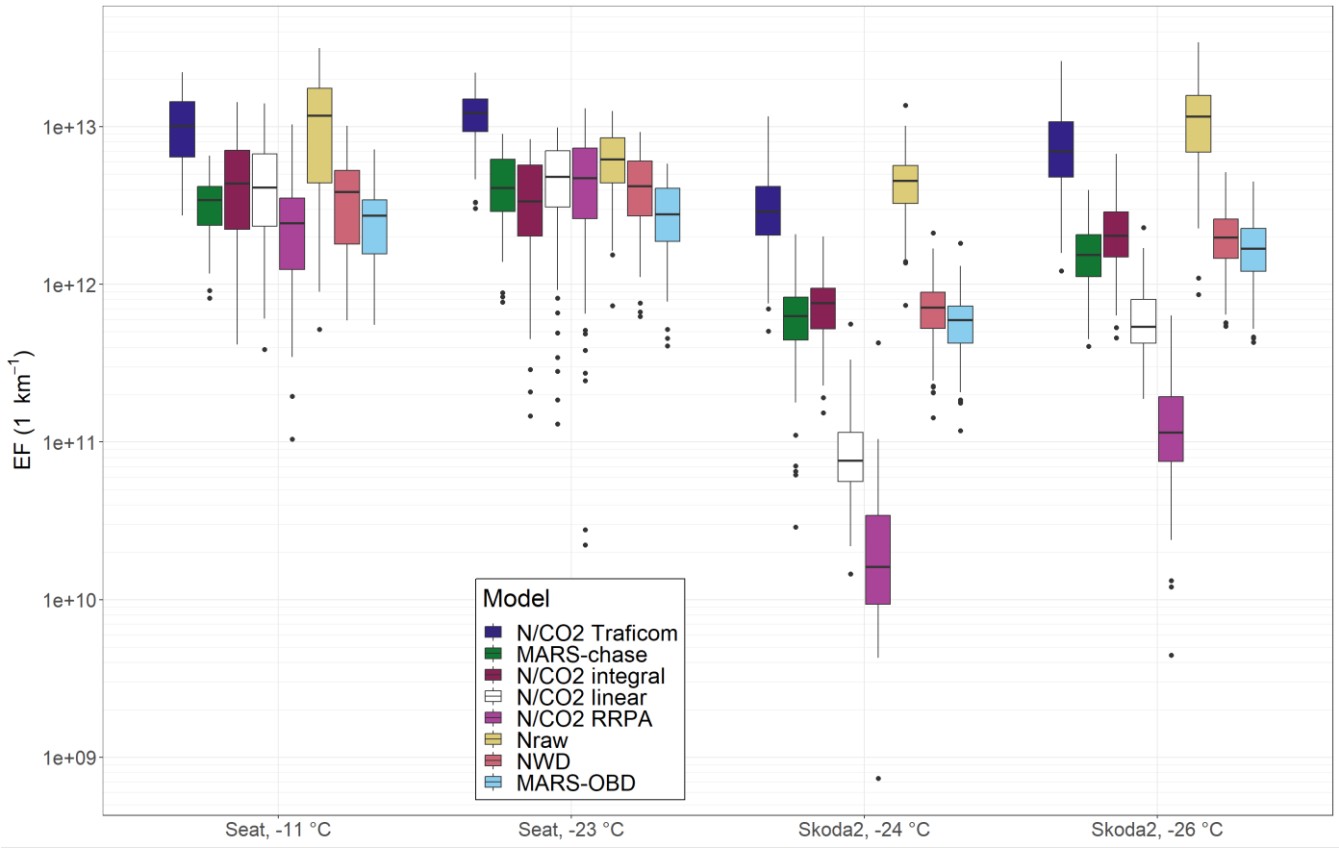

**Figure 7. Emission factors of > 23 nm particles for downhill sections and for Seat and Skoda 2, hot starts, except Seat –11 C that is with the subfreezing–cold start, and Skoda –26 C that is with the preheated–cold start. Results are calculated from 100 bootstrap samples (see Sect. 2.3 for the description of bootstrap sampling). For Skoda2, some EFs (89 for Skoda2, -24 °C and 39 for Skoda2, -**
**26 °C) are zero. Only EFs above zero are shown in this figure.**



## 4 Conclusions

There are methods to define DRs and EFs that do require OBD data from the vehicle under tests and methods that do not require. We conclude that most of the $N/CO_2$-methods are not suitable for transient driving, where EF is constantly changing during the drive, which is indicated by results that differ from the ones obtained with the other methods.

For those time points where the measured $CO_2$ is close to its background value, the new methods (the NWD and the MARS methods) work better than the old ones. Among these, the NWD method is physically more realistic and hence easier to interpret. We believe both the NWD, and the MARS methods introduced are extendable also to non-exhaust emissions. Both methods would require some prescribed database to characterize the effect of vehicle's shape on DR. The number of required vehicles for the database can be from one (if the interest is only emissions of a specific vehicle) to some hundreds of vehicles
(monitoring of emissions from random vehicles from the fleet).

The MARS methods are based on the dependencies of the measured variables on DR from the $N_{raw}$ method. It fixes the problems of the $N_{raw}$ method at the time points where DR is estimated to be very high with the $N_{raw}$ method. On the other hand, the MARS methods do not have as clear physical interpretation as the NWD method. The MARS methods are; however, very adaptive methods and DR could be modeled using variables other than the ones used in this paper, which might increase the
physical interpretability of the methods.

If the MARS-methods were used with other variables, for their generalizability, it would be beneficial to use such variables that are generally measured in the chase measurements. Positive sides of the MARS methods also include that in the MARS-chase method, no variables measured directly from the vehicle diagnostics are not needed. This enables the observation in the middle of traffic, also in driving situations where EF and DR cannot be assumed constant.

The weakness of these methods is that the time points with the vehicle speed of zero, have been omitted in this study. This limits the usability of the method in e.g., urban conditions where the vehicle is stationary a significant part of the time. In this study we wanted to focus especially on times when the vehicle is moving, including downhills, and the fuel flow rate is zero or close to zero. The times when the vehicle is stationary could be added to the methods (MARS and NWD) by separately considering the speeds of zero. In the first place, it could be implemented by using e.g., the $N_{raw}$-method for those times.

Vehicle chase studies in the future are not only limited to determination of the exhaust originated species, since the NWD method could be used to define the non-exhaust particle emission originating, e.g., from the brakes and tires of the vehicle under the test. In addition to being an important tool in emission research especially in real-world emission factor determination including the semi-volatile particles, the chase method has potential to be a monitoring tool for vehicle fleets in official purposes: high emitting vehicles could be identified while driving with simultaneous particle and $CO_2$ sensor signals and

processed for further detailed measurements according to e.g., the new PTI protocol where particle number concentrations are measured on low idle.

**Code and data availability**

Data will be opened at the time of final publication. Until the publication of the article, data can be requested from the corresponding author.

**Author contribution**

Ville Leinonen: Conceptualization, Data curation, Formal analysis, Investigation, Methodology, Writing – original draft preparation, Writing – review & editing

Miska Olin: Conceptualization, Data curation, Formal analysis, Investigation, Methodology, Project administration, Resources, Supervision, Writing – original draft preparation, Writing – review & editing

Sampsa Martikainen: Investigation, Writing – review & editing

Panu Karjalainen: Conceptualization, Funding acquisition, Investigation, Methodology, Project administration, Resources, Supervision, Writing – review & editing

Santtu Mikkonen: Conceptualization, Funding acquisition, Methodology, Project administration, Supervision, Writing – review & editing

**Competing interests**

The authors declare that they have no conflict of interest.

**Acknowledgements**

This research is a campaign of the "AHMA" project funded by the Jane and Aatos Erkko's Foundation, supported by the Academy of Finland project "EFFi" (grant no. 322120). P.K. acknowledges funding from Tampere Institute for Advanced
Study (Tampere IAS). S.Mi. is supported by the Academy of Finland competitive funding to strengthen university research profiles (PROFI) for the University of Eastern Finland (grant no. 325022). S.Ma acknowledges the funding from Kone Foundation. This research has received support from the Academy of Finland Flagship Programme "ACCC" (grant nos.



337550 and 337551). Tampere University's measurement van, ATMo-Lab, contributes to the INAR RI and ACTRIS infrastructures.

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
