# Peer review of "Challenges and solutions in determining dilution ratios and emission factors from chase measurements of passenger vehicles"

_Atmospheric Measurement Techniques, 2023_

## Author Comment (AC1)

We gratefully thank all reviewers for the careful reading and valuable comments. Below we provide our point-by-point responses to the reviewers' comments. In the following context, raised comments/suggestions are marked in **black**, responses are presented in **red**, and changes to the manuscript/supplement information are indicated in **blue**.

**Reply to Anonymous referee #3**

The paper deals with a specific issue of emission factor determination using plume chasing. Typically, the emission factors of the pollutants are determined by assuming dilution factors based on CO2 measurements. In special cases (e.g. downhill driving, engine motoring, hybrid operation) or for non-exhaust emissions, where CO2 emissions are low or non-existent, this approach fails. To overcome this problem, the authors present two methodologies, one based on a multivariate regression algorithm and a second model-based approach that takes into account speed, exhaust mass flow, vehicle shape, and fuel used. The methods have been tested against conventional methods using experimental data from a designated measurement campaign.

The methods developed in this work are important for remote emission sensing and address relevant scientific issues within the AMT, namely the determination of exhaust plume dilution factors also in the absence of CO2 emissions for individual vehicle emission factor measurements.

We thank the referee for these positive comments towards our manuscript.

However, the paper only deals with particulate emissions, which is not mentioned in the title or abstract.

We thank the reviewer for pointing this out. We added that this study deals with particulate emissions to two sentences in the abstract:

… "This problem is also encountered when studying non-exhaust *particulate* emissions, e.g., from electric vehicles." … "We show that emission factors for *particulate emissions* calculated with both methods are in line with the current methods with vehicles producing $CO_2$." …

It would be very interesting for the reader to know if the methods are also applicable to gaseous emissions and, if so, what the limitations are.

Even though this paper is focusing on particulate emissions, the dilution of the exhaust should be applicable also to gaseous variables. It needs to be assumed, as in this paper have been assumed, that $CO_2$ and particles/gaseous emissions are diluted similarly in turbulent exhaust plume. This is a common assumption also in earlier studies, such as Jayaratne et al. (2005) for particulate emissions.

It is mentioned, but not described, how the methods could be applied to non-exhaust emissions.

We added the following text to the conclusions section to describe our reasoning for that:

"For NWD, the method is based on the estimated slope $\kappa$ of the vehicle. For example, for tire emissions, if the emission from the tires is $C_{raw}$ and mass exhaust flow rate of the emission is $Q$, then $EF = C_{raw} * Q$. On the other hand, it was assumed that $DR = \kappa * v/Q$. Then $C_{raw} = C_{meas} * DR = C_{meas} * \kappa * v/Q$. For EF, we get that $EF = C_{raw} * Q = C_{meas} * \kappa * v$. Hence, an explicit value of mass exhaust flow rate $Q$ is not needed to calculate EF of non-exhaust emission. The $\kappa$ value can be estimated from the other vehicle with similar estimated dilution of emissions, or in case of hybrid vehicle, the $\kappa$ can be determined during the time when the combustion engine is running. For MARS the basic idea is that from the test dataset of measurements, the dilution ratio of emissions could be estimated in different driving situations. Then in the new dataset, the DR is estimated based on splines estimated from the test dataset.

In both methods, the emission factor of the non-exhaust emission can be determined during the times when the vehicle is running with electric engine only. For the non-exhaust emissions, some correcting coefficient for the dilution ratio might be needed."

In particular, it would be interesting to know whether the dilution of exhaust and non-exhaust emissions is comparable.

Dilution ratio of non-exhaust emissions can be different than for the exhaust emissions as the location of the emissions are different (e.g., brakes and tires vs. tailpipe). This should be considered in the emission factor calculations. If the DR for exhaust emissions is not valid for non-exhaust emissions, in most simple case, extra coefficient for non-exhaust emissions could be constant for all non-exhaust emissions, i.e. for dilution ratio $DR_{non-exhaust} = a_{non-exhaust} * DR_{exhaust}$. Other option would be to determine separate coefficients for different types of non-exhaust emissions, depending on the location of the emissions sources.

We added the sentence "For the non-exhaust emissions, some correcting coefficient for the dilution ratio might be needed." to conclusions section to highlight the possible differences in dilution ratios of exhaust and non-exhaust emissions.

It would also be interesting to know how the methodologies deal with hybrid operation of vehicles.

Please revise accordingly.

For chase without OBD data, hybrid vehicles are probably the hardest class of vehicles to estimate EFs, as the information about use of the exhaust engine and electric engine are not available. However, both methods determine the dilution of emissions on test dataset and hence the dilution ratio is not dependent on the operation of the hybrid vehicle. With OBD-data available, hybrid vehicles could be used to estimate non-exhaust emissions. The hybrid drive with using of electric engine only should produce only non-exhaust emissions. As mentioned in the comment about applicability of the methods to non-exhaust emissions, estimating the dilution of emissions in both methods should not be a problem for hybrid vehicles.

Minor:

Line 172: particle concentration: abbreviation N is missing

Corrected.

Line 204: typo: without no

Changed to 'without a need to'. In the same sentence, there was also 'determinethe'. That was corrected to be 'determine the'.

**References**

Jayaratne, E. R., Morawska, L., Ristovski, Z. D., and Johnson, G. R.: The use of carbon dioxide as a tracer in the determination of particle number emissions from heavy-duty diesel vehicles, Atmos. Environ., 39, 6812–6821, https://doi.org/10.1016/J.ATMOSENV.2005.07.060, 2005.

---

## Author Comment (AC2)

We gratefully thank all reviewers for the careful reading and valuable comments. Below we provide our point-by-point responses to the reviewers' comments. In the following context, raised comments/suggestions are marked in **black**, responses are presented in **red**, and changes to the manuscript/supplement information are indicated in **blue**.

**Reply to Anonymous referee #4**

The paper presents an interesting study where several methods for calculating the dilution ratio and the emission factor from a chased vehicle are tested and compared. It is an important contribution to the remote emission sensing research field, namely to the emission factor determination using the plume chasing technique.

We thank the referee for the positive comments towards our manuscript.

134 Could you estimate how good the assumption that the vehicle distance was constant between the two vehicles is since you had GPS on both the vehicle and the measurement station?

Good question. We tested this during the data preprocessing phase. We found that the GPS devices we used for measuring location were too inaccurate for measuring the distance between two vehicles.

As an example, we had a stop at the same location (+-1 meter from the certain sign). Before the stop, the distance between vehicles was recorded to be between 1 and 10 meters (5 and 95% quantile) and during the stop, the distance was between 3 and 12 meters (5 and 95% quantile). There was no clear difference in distance between vehicles. Especially during the stop, the distance was standard, and that could be checked from the video material. Hence the 10-meter difference between the rounds is at least mostly caused by the measurement accuracy of GPS devices.

For that reason, we assumed that the distance was constant during measurement period. Probably, the distance between the chased car and mobile laboratory in chase measurements should be recorded better in the future.

We added a sentence to the text to clarify this ("Unfortunately, the GPS data from the chased vehicle and ATMo-Lab was not accurate enough, so that the changes in the chase distance could have been estimated from the GPS data.").

In line 220 you state that Nraw uses $CO_2$ calculated from OBD data while in Table 2 it is in a row stating it is not using OBD data. Can you make this consistent?

$N_{raw}$ method is using OBD data as the information of the exhaust flow rate is measured by OBD device. As the Table was placed so that it continued the next page, it was not clear to which category $N_{raw}$ belongs. To make it clearer, we moved the whole Table to the next page, and inserted one empty line below $N_{raw}$.

**Table 2: Division of the methods for calculating EF of a vehicle. OBD data means the data collected from the chased vehicle (see also Figure 2) and learning data means the data collected from other drives of the same vehicle and from other vehicles (including data from ATMo-Lab and, also from OBD if its data is used). Methods are introduced in more detail in subsections 2.3.1-2.3.7.**

|  |  | Uses learning data | |
|  |  | yes | no |
|  |  |  | $N/CO_2$ integral, |
|  |  |  | $N/CO_2$ linear, |
| Uses | yes | MARS-OBD, | $N/CO_2$ RRPA, |
| OBD |  | NWD | $N_{raw}$ |
| data |  |  |  |
|  | no | MARS-chase | $N/CO_2$ Traficom |

Is Audi type-approved as Euro 5? Euro 5 type approval started in September 2009, while Audi's registration year was 2008. Where is the information on the studied vehicles from?

The information of the studied vehicles is from the Traficom database. The information from the Traficom database states that Audi fulfills Euro 5 requirements.

186 – 193 In the N/CO2 Traficom description you say you used the Traficom data, can you also state how the values you got there compared to what you measured and what the producer of the vehicle states the values are?

Traficom provided consumption values for each vehicle. Traficom consumption values are based on the values provided by the manufacturer of the vehicles.

The consumption value used here is so called "combined consumption" which combines the urban and road driving consumption, the two other values Traficom provides. This was selected because it was the only value that was provided for all vehicles. In addition, it is most likely the most optimal out of those three for our driving route.

Table Y below shows the values from Traficom database, compared to measured (OBD) consumptions of the vehicles. We can see that usually the measured values were on average higher than the values provided by Traficom. Of course, for certain sections such as the downhill section, the fuel consumption can be much lower than the median value. Cold outside temperatures are at least one reason for larger median consumption but also the temperature of the engine might have an effect, especially at the beginning of the drive.

Table Y. Measured fuel consumptions and reported combined fuel consumptions from Traficom database.

| Vehicle | Median measured fuel consumption (l/km) | Reported combined fuel consumption (l/km) |
|---|---|---|
| Audi | 9,9 | 7,1 |
| Seat | 6,7 | 5,7 |
| VW | 8,4 | 7,6 |
| Ford | 7,1 | 4,6 |
| Skoda1 | 6,8 | 5,4 |
| Skoda2 | 7,4 | 7,5 |

We added the following sentences to subsection 2.3.2 to clarify the origin of Traficom consumption values and the nature of those values, as an average fuel consumption.

"Traficom consumption values are based on the values provided by the manufacturer of the vehicles. The values in the database are the average consumptions (in unit of $l/100\ km$), and hence the actual consumption at certain time might be over or under the consumption value in the database."

195-213 One-minute constant EF seems long. Looking at Figure 1, could you use a shorter time maybe 30s? How long did it take you to make a single round of 13.8 km? Was there much variability between runs?  Would this affect the result? How are the zero values included in the final EF distributions since these are on log plots?

The drive time was between 19 min 41 sec and 22 min and 42 sec, including two 30 second stops and two 1 minute idles before and after the drive. However, there is variation between vehicles and within drives of one vehicle. See boxplots for ranges of drive times for each car in the figure 2 a) below.

EFs were calculated for the shorter time periods and then pooled to get an EF for the whole round. We didn't find any clear dependence between drive time and EF. See also Figure 2 b) below for median EFs from N to CO2 ratio linear method calculated as in Figure 5 for all the vehicles.

[Figure]

Figure 2. a) Boxplot of drive times for each vehicle. b) Scatter plot of EF from N to CO2 linear method (y-axis) as a function of drive time (x-axis).

Zero values were excluded from the EF plot in Figure 7. This has been mentioned in the figure caption. In Figures 5 and 6 all EFs were positive.

What values do whiskers on box plots represent (Figures 5 and 6)?

EFs were calculated to 100 bootstrap samples. For each sample we got an EF value, and those were represented as a boxplot.

We added a sentence "Whiskers are representing the distribution of EFs in different bootstrap samples." to both figure captions to clarify this.

Figure 4 is not clear – which splines are overlapping? Are all vehicles on all plots? It is difficult to see which vehicle is which with this color code.

We tried to make those splines more visible by removing the points from the figure and by changing the line type. See the new figure below this text. The overlapping splines are mentioned in the figure caption of Figure 4 but are hopefully now also easier to recognize also from the figure.

[Figure]

Maybe add a time resolution column to Table 3. Would different time resolutions influence your results?

We added the clarification to Table 3, for N/CO2 linear and RRPA methods using one-minute time intervals for linear models. Otherwise, all methods are using data with 5 second time resolution.

When using 5 second time resolution, we felt that 30 second intervals might be too short for linear fitting, as maximum of six points are measured in each 30 second period. For time resolution differences, we tested different time resolutions for linear and RRPA methods (30 sec, 1 minute). The results were slightly different when using different time resolutions, but the conclusions were not changed.

**Minor issues**

Use subscripts consistently throughout the text for "2" in $CO_2$ (there are two cases in Table 3 and in l. 175). Also degrees for degrees C in the caption of Figure 5.

We thank the reviewer for noticing these. These have now been corrected.

204 "without no need to" do you mean double negative or "with no need to" or "without a need to"

Changed to 'without a need to'.

---

## Author Comment (AC3)

We gratefully thank all reviewers for the careful reading and valuable comments. Below we provide our point-by-point responses to the reviewers' comments. In the following context, raised comments/suggestions are marked in **black**, responses are presented in **red**, and changes to the manuscript/supplement information are indicated in **blue**.

**Reply to Anonymous referee #1**

The paper presented by Leinonen et al. summarizes different methods to determine dilution ratios and emission factors from vehicle chase studies. The study location was a well-suited test road for wintertime investigation. Mostly, no other vehicle intervened the setup. The study appears to be scientifically sound and adds valuable data to the literature of exhaust chasing. It should be accepted for publication after some revision.

We thank the referee for these positive comments towards our manuscript.

The study would have been much more useful and credible if the test vehicle would have been equipped with a PN-PEMS. This would have proven if the order of magnitude for PN of the Diesel vehicle (assumed with DPF) is correct. An average of 5x10E12 #/km at -11C to -23C could have easily been confirmed, or disproven. The same question arises for the gasoline vehicle: 4x10E11 – 1x10E12 #/km (Skoda-24C to -26C ). How does this compare to PN PEMS during the same drive?

The referee is correct that PEMS could have been used to compare the emission factors of vehicles. In earlier study, such as (Karjalainen et al., 2014) have shown that the emissions of particles measured from the chase measurements are in line with the dynamometer results. Based on those earlier results, we believe that the emission factors calculated in this study are in line with the results that would have been achieved with the PN-PEMS system. In the future, the comparison between PN-PEMS and chase measurement methods would be interesting to conduct.

The results for the Skoda-2 -24C/-26C selecting downhill driving are puzzling: Why is the EF not substantially lower (downhill Figure 7) if compared to the overall trip? Rather the opposite occurs comparing Skoda -24C Figure 5 (about 4x10E11 #/km) with downhill Figure 7 (about 6x10E11 #/km)?

[Figure]

Figure 1. Time series of $CO2_{meas} - CO2_{bg}$ (black) and $N_{meas} - N_{bg}$ (blue) for the downhill section for Skoda2, -24 C round. Altitude profile is shown with gray background ribbon. Note that the data between 10:20:00 and 10:21:00 is not used in the calculation of downhill EF:s.

For the high EF of Skoda2 -24 C round, Fig. 1 shows the time series $CO2_{meas} - CO2_{bg}$ and $N_{meas} - N_{bg}$ for the downhill section. We also checked the video recording for the downhill sections. For the first downhill, between 10:19 and 10:20, there was no clear braking period. Hence the sources of particles measured from that time period were not clearly identified. For the second downhill, after 10:21, there were multiple breakings starting from 10:21:25 and 10:21:55, which is close to the increase of number concentrations observed. However, the chemical composition of those particles was not studied in detail for this study. Those concentrations shown in figure 1 explain the high EF (in #/km) shown for the downhill section in Figure 7.

What were the chasing distances and can could the NWD, or MARS be applied in open traffic, where this short distance would be potentially unsafe?

When the chased vehicle was moving, the chasing distance was between 5 and 10 meters. So short distances won't be safe in open traffic, especially under winter conditions, if the chased vehicle is driven by a random driver. Chasing methods, however, could be used for other (longer) distances as well, as shown e.g. in Olin et al. (2023).

We added the information about estimated distance between vehicles into section 2.3.

"Based on our estimation, the chasing distance was between 5 and 10 meters when the chased vehicle was moving."

The authors are measuring total PN, and no size distributions. Therefore, it is left entirely open if nucleation particles would have occurred at the low temperatures. If nucleation would have occurred, if would question the method of NWD, particle formation could not have been completed in the exhaust plume.

Maybe, for a next study a setup w and w/o thermal treatment of the PN will be included?

It is difficult to estimate whether the nucleation and condensation processes are complete prior to the sampling. The plume residence time was at minimum about 0.5 s. The focus of the article is rather on different computational methods to define the dilution and emission factor, not to cover the fine details of aerosol particles and properties. More thorough aerosol characterization study for sure would require the use of thermal treatment and measurement of different particle sizes.

Other editorial:

Line 23: please clarify what is meant with "but the regulation even for new vehicles is still under development and the new regulations do not completely cover the existing fleet"? If a future regulation is meant, it is normal. It has to be developed and can only apply to vehicles coming into production at that date.

We agree with the reviewer that the meaning of the sentence was not clear. The sentence was modified to "Vehicle emissions are regulated in legislation but the regulation for new vehicles is under constant development (type approval, periodical technical inspection (PTI), and real driving emissions (RDE)). The new and upcoming regulations are effective only for the vehicles produced after the regulation have become effective."

Line 28-30: "The limits for PN only consider nonvolatile particles, and the particle mass (PM) formed from the precursor gases via nucleation and condensation as the exhaust gas dilutes and cools upon exiting the tailpipe is mostly neglected. The amount of particle matter (both in terms of PN and PM) formed this way can be considerable."

These sentences are misleading: With the elimination of fuel sulfur the occurrence of volatile PN formed upon cooling and nucleation has been decreased. The hydrocarbons potentially nucleating are measured and regulated via gaseous HC requirements. Also, potential health effects are more likely to be related to solid PN (i.e. more long-lived, and not dissolved and diluted in the alveoli).

Therefore, until data is available, or a strong reference is added, the sense of the statement "neglected" should be revised.

We clarified the message based on the comments from the reviewer. The updated version of the sentences is as follows:

"The regulation limits for PN mostly considers nonvolatile particles. The particle mass (PM) formed from the precursor gases via nucleation and condensation as the exhaust gas dilutes and cools upon exiting the tailpipe is not fully considered PN measurements, however the regulation for gaseous hydrocarbons limits the amount of precursor gases produced by the vehicle. The amount of

secondary particle matter (both in terms of PN and PM) formed from precursor gases can be considerable. However, the amount of secondary PM has decreased in 21$^{st}$ century as the fuel does not contain as much sulfur as before."

Tab 1: Add if particle filter was installed. Euro-5 should have DPF, for gasoline vehicles not clear.

We added a column to Table 1 indicating filtering technologies in each vehicle. Particle filter was in every vehicle, except Ford Focus. An updated version of Table 1 is presented below.

Table 1: Information on the studied vehicles. DPF = diesel particle filter, GPF = gasoline particle filter, MHEV = mild hybrid electric vehicle, SCR = selective catalytic reduction

| Car | Fuel | Filter | Registration year | Engine displacement (l) | Emission class | Odometer reading (km) | Number of drives |
|---|---|---|---|---|---|---|---|
| Audi A6 | Diesel | DPF | 2008 | 3.0 | Euro 5 | 236,000 | 6 |
| Seat Alhambra | Diesel | DPF + SCR | 2012 | 2.0 | Euro 5 | 169,000 | 6 |
| VW Transporter | Diesel | DPF + SCR | 2019 | 2.0 | Euro 6 | 36,000 | 4 |
| Ford Focus | Gasoline | | 2018 | 1.0 | Euro 6 | 78,000 | 5 |
| Skoda Octavia 1.0 | Gasoline (MHEV) | GPF | 2020 | 1.0 | Euro 6 | 1,000 | 6 |
| Skoda Octavia 2.0 | Gasoline | GPF | 2019 | 2.0 | Euro 6 | 21,000 | 6 |

Line 377: "NWD and MARS Method would be extendable to non-exhaust emissions"

This statement should be further explained. How to differentiate from Exhaust PM? How to differentiate tire wear, and brake wear? Applicable to electric vehicles only?

We agree with the reviewer that the distinction of non-exhaust emissions from exhaust emissions might not be easy, and that it is probably easiest to study non exhaust emissions from electric vehicles. The methods presented did not differentiate any emissions from each other, as the methods focus on the dilution of emissions. The differentiation, if required, needs to be done using some other data/methodology. Additionally, the differences in dilution of emissions, i.e. exhaust vs. non-exhaust and also different sources of non-exhaust, needed to be considered.

We added the following text after the sentence:

"For NWD, the method is based on the estimated slope $\kappa$ of the vehicle. For example, for tire emissions, if the emission from the tires is $C_{raw}$ and mass exhaust flow rate of the emission is $Q$,

then $EF = C_{raw} * Q$. On the other hand, it was assumed that $DR = \kappa * v/Q$. Then $C_{raw} = C_{meas} * DR = C_{meas} * \kappa * v/Q$. For EF, we get that $EF = C_{raw} * Q = C_{meas} * \kappa * v$. Hence, an explicit value of mass exhaust flow rate $Q$ is not needed to calculate EF of non-exhaust emission. The $\kappa$ value can be estimated from the other vehicle with similar estimated dilution of emissions, or in case of hybrid vehicle, the $\kappa$ can be determined during the time when the combustion engine is running. For MARS the basic idea is that from the test dataset of measurements, the dilution ratio of emissions could be estimated in different driving situations. Then in the new dataset, the DR is estimated based on splines estimated from the test dataset.

In both methods, the emission factor of the non-exhaust emission can be determined during the times when the vehicle is running with electric engine only. For the non-exhaust emissions, some correcting coefficient for the dilution ratio might be needed."

**References**

Karjalainen, P., Pirjola, L., Heikkilä, J., Lähde, T., Tzamkiozis, T., Ntziachristos, L., Keskinen, J., and Rönkkö, T.: Exhaust particles of modern gasoline vehicles: A laboratory and an on-road study, Atmos. Environ., 97, 262–270, https://doi.org/10.1016/j.atmosenv.2014.08.025, 2014.

Olin, M., Oikarinen, H., Marjanen, P., Mikkonen, S., and Karjalainen, P.: High Particle Number Emissions Determined with Robust Regression Plume Analysis (RRPA) from Hundreds of Vehicle Chases, Environ. Sci. Technol., https://doi.org/10.1021/ACS.EST.2C08198, 2023.